# Exploring the Sustainable Delivery of Building Projects in Developing Countries: A PLS-SEM Approach

**Ahmed Farouk Kineber** [1,*] and **Mohammed Magdy Hamed** [2,3]

1   Department of Civil Engineering, College of Engineering in Al-Kharj, Prince Sattam Bin Abdulaziz University, Al-Kharj 11942, Saudi Arabia

2   Construction and Building Engineering Department, College of Engineering and Technology, Arab Academy for Science, Technology and Maritime Transport (AASTMT), B 2401 Smart Village, Giza 12577, Egypt

3   Department of Water and Environmental Engineering, School of Civil Engineering, Faculty of Engineering, Universiti Teknologi Malaysia (UTM), Skudia 81310, Johor, Malaysia

*   Correspondence: a.farouk.kineber@gmail.com or a.kineber@psau.edu.sa

**Abstract:** Sustainable building approaches should be included at every level of the development process for maximum benefit without jeopardising the structure's intended usage. However, researchers have paid less attention to how they may be applied to developing nations. This study aims to investigate the numerous determinants of sustainable delivery in the construction industries of developing nations. For this to happen, existing literature was used to inform the development of a closed-ended questionnaire. Consequently, 95 structured questionnaires by building professionals investigated the importance of these factors. As a result, the factor's structure was determined and confirmed using the study's partial least square structural equation modelling (PLS-SEM) approach, which was utilised to investigate the connections between these factors. According to the PLS-SEM analysis, the factors most strongly influencing sustainable delivery are those associated with the preparation, followed by evaluation and use factors, respectively. As a result of this research, authorities in Egypt's building sector will have a road map for implementing sustainability principles to reduce building costs, boost the local ecosystem, and strengthen social cohesion.

**Keywords:** building sector; Egypt; PLS-SEM; sustainability

## 1. Introduction

The residential development sector is often a barometer of the well-being and contentment of a nation's people [1]. About 40% of the energy used worldwide and 30% of greenhouse gas emissions come from buildings used as homes [2]. The supply of homes is inadequate to meet the growing demand in today's increasingly urbanising globe [3]. Consequently, increasing urbanisation makes it more difficult for low-income people in developing and industrialised nations to obtain affordable housing [4]. As many as 828 million people in extreme poverty in developing nations live in slums or substandard homes. These estimates suggest that by 2020, the figure might have risen to a staggering 1.4 billion [3,5,6]. The home's value in maintaining subsistence in these expanding cities is evident [7]. As a result, several pieces of affordable housing legislation have been passed by national governments, making affordable home building a national priority [1]. However, there is also disagreement about whether modest incomes can afford apartment buildings [3].

Sustainable development planning is integral to every building project manager's toolkit [8]. Although sustainability theory and practice are well-represented in the current literature, researchers have paid less attention to how they may be applied to the project management methods employed in less developed nations [9]. Researchers have made strides toward incorporating sustainability considerations into the administration of building projects [10]. The construction industry accounts for around 41% of global energy

consumption and 40% of global carbon dioxide ($CO_2$) emissions, so this information gap raises serious concerns [11].

Key project management team members must be aware of, and work to eliminate, any barriers to the widespread use of sustainable construction approaches [10,12]. For this reason, project management is essential in the construction industry [13]. While considerable progress has been made in incorporating sustainability into construction project management, there are still several significant gaps in the literature [9]. To begin, while examining stakeholders' viewpoints on sustainability adoption, the opinions of building professionals have been the most thoroughly investigated [14,15]. Second, researchers tend to assume that the industry will be eager to accept sustainability goals [16], even if there are reasons to believe this will not be the case. This contradicts what we know about construction workers, who are notorious for their inherent reluctance to change and ambivalence [17]. Finally, a more all-encompassing, comprehensive approach is occasionally overlooked in favour of an emphasis on a specific dimension or phenomenon (such as the role of the project manager [10] or the planning phase of the project [18]). Given the dearth of cross-disciplinary literature on sustainability in construction project management, this investigation is necessary, especially in a developing country such as Egypt. The literature has therefore highlighted that "sustainable buildings" must be built in a way that is both environmentally friendly and resource-efficient [19]. Wolstenholme et al. [20] expand on this idea by calling for a fundamental change in the way buildings are constructed, one that prioritises efficiency and sustainability. Experts in the building industry cannot precisely gauge environmental effects during the design phase of a project [21].

These considerations lead us to suggest the following question for the empirical study. What factors affect the efficiency and affordability of building projects in developing nations? This paper is the first attempt to draw attention to these elements. Scholars will profit from the results since they will provide insight into the nature of these elements and open the door to further research in this area. As a result of having this list at their disposal, practitioners and policymakers will be better equipped to put these considerations into practice and will be able to zero in on the areas that need the most work to become sustainable.

## 2. Research Background

### 2.1. Incorporating Sustainability into the Construction Industry

For long-term economic success, scholars worldwide have stressed the importance of integrating sustainability into building project management [22–25]. New evidence reveals that rising incomes, a growing consumerist culture, rapid population increase, and unchecked urban development all contribute to the negative environmental effect of the building sector [25–28]. It is possible to achieve environmental, economic, and social goals in a sustainable macroeconomic environment [25,29,30].

As applied to the microeconomics of construction project management, sustainability ensures that a constructed environment is adequately maintained throughout an asset's useful life while also being environmentally friendly [8]. The findings of recent studies on the environmental effect of construction activities have been noticed by international industry actors (including governments, building professionals, scientific communities, businessmen, and clients) [25,31]. Since it contributes to and benefits from the production and consumption of raw materials and completed items in the supply chain, this sector has a disproportionately significant impact on the environment [32].

Waste generated during the building process accounts for roughly 30–35% of construction expenditures [33,34], and the sector employs more than 111 million people worldwide [35]. Additionally, the building sector accounts for nearly 40% of global energy consumption and 40% of greenhouse gas emissions [35,36]. Sustainable construction is a high priority for governments in the developed world, while in the underdeveloped world, economic growth is still given more priority [14]. The expanding importance of

economic growth for attaining social equity [14,37] has pushed environmental problems to the background as the necessity for construction has increased in emerging nations.

Numerous studies have either looked at the methods used to undertake sustainable development activities [38,39] or tried to include sustainable activities in construction operations [18,40]. It was never investigated whether or not a smooth changeover is achievable [37]. The current study hypothesises that a more practical approach to implementing sustainability in construction projects for underdeveloped countries is to identify carefully, prioritised analyses of the existing impediments [14]. Sustainability may significantly impact how a project is delivered at each stage of its life cycle [10]. Therefore, it is necessary to consider the whole project life cycle if the sustainability attitude is to be fully adopted and ingrained into current practice.

### 2.2. Factors Affecting the Sustainable Delivery of Building Projects

For a construction project to be sustainable, it must serve many often conflicting goals, including environmental protection, social betterment, and the advancement of the construction company's strategic interests and financial well-being [41]. This implies that sustainable practices will be included in every project development and implementation stage [42,43]. Keeping this in mind, the current body of research on sustainability may be broken down into a few distinct groups. Standards [44], sustainable resources [43], eco-design, and corporate responsibility are only a few of the topics that have been researched in the realm of construction [41]. Stakeholder management [45], lifecycle management [46], and sustainability assessment are all examples of related topics that arise throughout the delivery phase of a project [47,48]. Sustainable project organisations [49], sustainable project practises [14], and sustainable project management [49] have also been the subject of research. Those developing nations with a penchant for sustainability have often adopted Western techniques [14,50,51].

The triple bottom line (TBL) is a widely used framework for understanding sustainable development. It considers not just economic but also social and environmental (or ecological) considerations (or financial). People, planet, and profit are sometimes called the "three Ps". As defined by Willard [52], these three elements make up the foundation of sustainable development. Construction sector organisations have been driven to adopt the concept of sustainability in building projects due to the growing trend toward clean production, green products, and increased awareness of climate change [42,53]. Social and environmental advancement, on the one hand, and the construction company's strategic interests and bottom line, on the other, are often at odds with one another, making it difficult to achieve sustainability in building projects [41]. Taking this approach entails incorporating sustainability ideas into the project's actions from the beginning to the end [42,43]. Keeping this in mind, there are numerous groups into which the existing body of sustainability research can be sorted [12]. Standardisation research [44], sustainable materials research [43], eco-design research, and CSR research are all examples of from the building sector [41]. Stakeholder management [45], lifecycle management [46], and sustainability assessment are three examples of such groups involved in the actual delivery of projects [48,54]. There have also been studies into sustainable project management [49], sustainable project organisations [55], sustainable project practices [14], and sustainable decision-making [55]. Countries in the developing world that are making efforts toward sustainability often model their policies and procedures after those of more industrialised nations [14,50,51]. However, the criteria that determine the sustainability of construction projects are extremely context-specific, and the methods used in advanced nations may not be applicable in less developed nations [14]. So far, there has been a less academic inquiry into how sustainability principles could interface with building initiatives in Egypt. Most of the ones done so far have taken an exploratory approach, focusing on certain subfields of sustainability. Though limited, the results do indicate delivery issues in underdeveloped nations [56]. Environmentally friendly, resource-efficient "sustainable buildings" have been called for in the literature [19]. In addition, by 2030, the Egyptian government hopes to

have Egypt ranked among the top 30 countries in the world [57]. As a result, sustainable practices should be incorporated into Egyptian construction projects [58]. It is important to note that the factors that affect the long-term viability of construction projects vary greatly depending on their location; as a result, practices common in more developed countries may not be appropriate in less economically stable regions [14]. Table 1 presents the most relevant parameters that determine sustainable delivery for building projects, based on the study of Hosseini et al. [56].

**Table 1.** Factors affecting sustainability in construction projects.

| Code | Factors Leading to Sustainability | Studies |
|:---:|:---:|:---:|
| | **Evaluation** | |
| E1 | Establishing a reliable system for strategic planning | [59] |
| E2 | Stakeholders' firm dedication to the project's long-term success | [60] |
| E3 | Respect for the interests of parties other than the client | [61] |
| E4 | Consistency in the use of anti-corruption policies throughout the decision-making | [62] |
| E5 | To develop sustainability principles in megaprojects, governments and professional organisations must enact necessary policies | [63] |
| E6 | All parties involved have agreed upon and articulated their top priorities | [64] |
| E7 | Sustainable project outcomes that are in line with stakeholder priorities | [65] |
| E8 | Clear goals and boundaries for the project | [62] |
| | **Preparation** | |
| P1 | Insight into PMT's understanding of sustainable project delivery | [60] |
| P2 | Positive interactions between project participants predominate | [62] |
| P3 | Comprehensive contract and specification documentation | [59] |
| P4 | Effective prerendering and tendering investigations | [66] |
| P5 | Formation of PMTs based on expertise and openness | [66] |
| P6 | Defining duties, responsibilities, and authority inside a company | [65] |
| P7 | In-depth research of the contractors' past work to determine their familiarity with the notion of sustainability and their track records in implementing sustainable projects | [60] |
| | **Use** | |
| U1 | Emphasis on high-quality workmanship | [65] |
| U2 | The positive reception from the public | [66] |
| U3 | Security in the economy and government | [67] |
| U4 | Supportive organisational norms for long-term project success | [68] |
| U5 | The skill and knowledge of project managers | [64] |
| U6 | Consistent access to all necessary resources (money, equipment, supplies, etc.) throughout the project's duration | [69] |
| U7 | The transparent and competitive procurement process | [66] |

## 3. Methods of Research and Model Construction

The drive of this study is to catalogue and evaluate Sustainable Building Project Delivery in Egypt. Following a thorough analysis of the available literature, 22 contributors to long-term implementation were recognised, as demonstrated in Table 1. The questionnaire poll was conducted by sending out a list of obstacles to home construction industry professionals. This was done so that the aspects of implementing sustainability would be as clear

and comprehensive as possible. For this reason, the methodology is followed in Figure 1. Adopted from Kineber et al. [58].

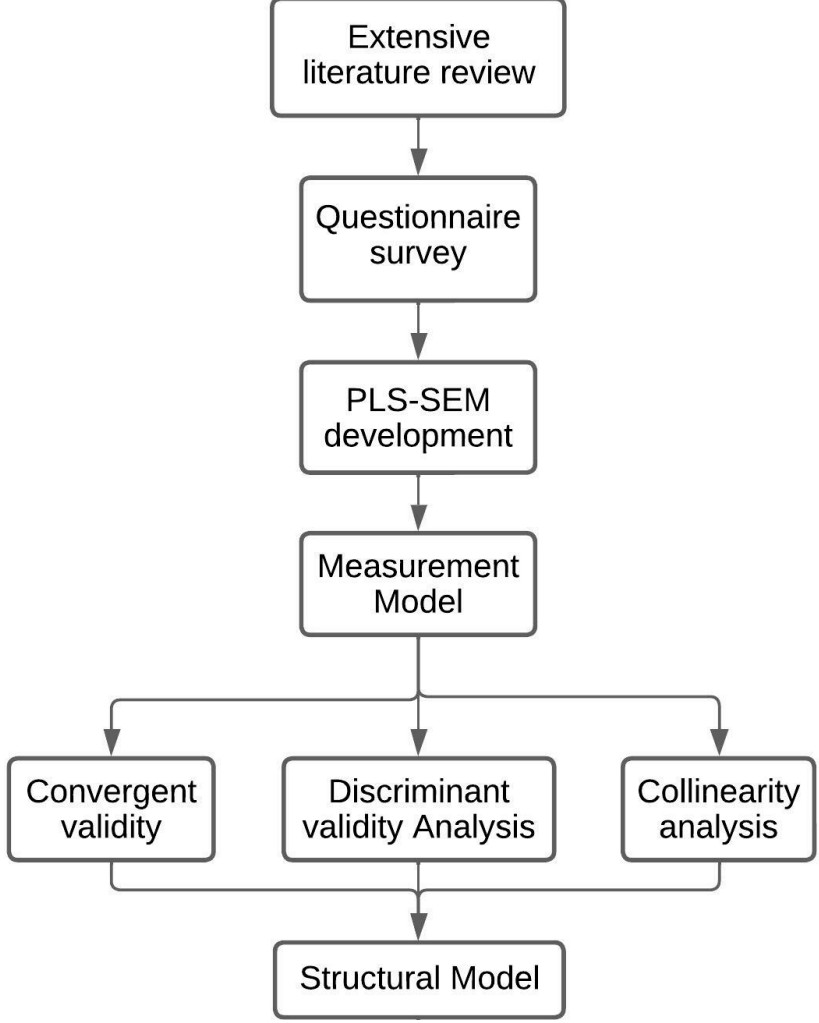

**Figure 1.** Research design.

### 3.1. Defining the Factors

Different steps of data collecting, refinement, and categorisation have been used, and the literature review technique has been investigated through (exploratory) research [70]. Articles, research papers, government documents, green construction codes, and other materials have all been carefully considered to compile the data for this study. Information is whittled down in the review process by being chosen, simplified, and abstracted. This refers to organising information into broad concepts [71]. Hosseini et al. [56] categorisation was used as a basis for this classification, and care was taken to assign relevant material to the appropriate subgroups (such as subthemes) within the main categories [70]. We broke down the whole thing into global, regional, and country categories, with a focus on developing nations. Plus, the most important concerns are raised by construction projects.

### 3.2. Pilot Survey

After consulting with executives, market researchers, and academics, the initial set of questionnaire questions was drafted. To guarantee the reliability of potential data-gathering techniques, a Pilot test method was incorporated on a modest scale [72]. A pilot study is conducted to test the research design methodologies and see if the procedures and methods used in the full study are adequate [72]. Workmates, experts, and superiors all provided

feedback that led to revisions to the questions. In most cases, questions on the questionnaire were reworded as part of the instrument's revisions.

### 3.3. Questionnaire Survey

A larger range of possible participants in Egypt's residential building sector was recruited for a questionnaire study to learn more about the barriers to sustainability implementation in Egypt. The demographics of the respondents, the barriers to implementing sustainability, and the perspectives of the respondents themselves made up the three parts of this study (Table 1). The questionnaire was made to allow for free-form responses (to add any barriers that the participants considered essential). Those involved in the process were notified, including contractors, consultants, and customers. These professionals include architects, electrical engineers, quantity surveyors, and structural and mechanical engineers. Based on their understanding and observations, respondents assigned numerical values to these aspects: 5 for very high, 4 for high, 3 for normal, 2 for small, and 1 for nil or very little. Previous research has used this scale [58,73–77].

### 3.4. Population and Sampling Method

Considering the novelty of the concept of sustainability in Egypt, stratified sampling was used to investigate its effects on a subset of the population. In addition, the sample size used in this study was decided by a purpose analysis of the methodology [78]. In total, 95 out of 150 people were contacted for this study, with a response rate of around 70%, by Yin [79] recommendation that a small sample size is ideal for doing PLS-SEM. According to the findings of these studies, this rate of return is satisfactory [80,81].

### 3.5. Validity and Reliability Analysis

Quality and efficiency in the analysis are of utmost importance, so it is important to have a well-thought-out theory framework, as well as acceptable protocols and methodologies, before beginning any investigation. Research findings are trusted because of reliability and validity, as stated by Neuman [82]. Testing for reliability is often performed to rule out biases and flaws in the analysis and guarantee that the same results will be achieved if the researcher repeats the same technique. This contrasts with validity, which is based on objective evaluations of the study's quality and approximation of the truth. The method of data collection was determined after reliability testing to guarantee repeatable results. However, face validity and content validity are the two conventional measures of validity used to evaluate whether the components of the survey instrument are appropriate and reflective of the intended research design [83]. To ensure the reliability of the research instrument, a sample of 12 research professionals from the field and the academy were chosen at random. This was done by the suggestion made by Sushil and Verma [84], that face validity is checked by having experts assess the contents to test them and ensure they appear sufficient. Experts analyse the substance of the test to ensure the items seem adequate (face validity). To determine the validity and reliability of the findings, Cronbach's alpha was utilised. The reliability of the scale was assessed by examining the correlations among the items in the sample to determine whether they were indeed related. Values of Cronbach's alpha above 0.8 are considered very reliable, while values between 0.6 and 0.7 are considered adequate. Since each of the alpha Cronbach values is more than that, this means that they are all suitable. Six, the average set correlation was greater than 0.3, suggesting that there were stable internal variables present in all objects [85].

### 3.6. Analytical Approach (PLS-SEM)

The study's fundamental purpose is to investigate what elements contribute to long-term sustainability in delivery using SEM as a forecasting tool. To meet the needs of the research, the SEM technique describes the relationship between as many observable and unobservable variables as necessary. According to Amaratunga et al. [86], SEM is an effective method for addressing the presence of errors in variables. The study used the structural

equation modelling technique to create a model and determine the connection between elements related to sustainable delivery. To fill the gap left by the lack of familiarity with hypothesis testing procedures, Byrne [87] argued that structural equation modelling (SEM) has become a standard non-experimental research tool. Additionally, using research articles from the MIS Quarterly, Ringle et al. [88] concluded that this technique has improved with time. Yuan et al. [89] also found that SEM is a common and widely used method for analysing social data. The SEM method was utilised in the study since it is widely known to be effective in the construction industry [90]. Additionally, the activities in the sustainable delivery elements have been evaluated using a PLS model that includes both reflective and formative variables.

## 4. Analysis and Findings

### 4.1. Respondents' Characteristics and Demographic Profiles

As shown in Table 2, this study looked at participants' comprehension and awareness of sustainable construction. According to the results, respondents were "cquainted" and "completely familiar" with 46.9% and 24.5% of the sample, respectively. The results also revealed that "Professionality" emphasised that "Architect" (12.2%) and "Civil Engineer" (66.3%) were seen to have the highest frequency and "Electrical Engineer" (4.4%) to have the lowest frequency, respectively. For bachelor's, master's, and doctoral dgrees, the qualifications of the respondents were measured at 65.3%, 13.3%, and 18.4%, respectively. Table 2 also reveals that about 52% of respondents had a job for between one year and five years or less. Approximately 27.6%, 2%, and 6.1% of respondents had job experience ranging from 5 to 10 years, 11 to 15 years, and more than 25 years, respectively. This suggests that the study's participants are knowledgeable and able to learn from it.

**Table 2.** Demographic analysis.

| Level of Awareness in Sustainable Construction % | |
|---|---|
| Totally Familiar | 24.5% |
| Familiar | 46.9% |
| Moderately Familiar | 23.5% |
| Not Familiar | 5.1% |
| **Highest level of education qualification %** | |
| Bachelor's degree | 65.3% |
| Diploma | 3.1% |
| Master's degree | 13.3% |
| PhD | 18.4% |
| **Years of experience in line construction projects %** | |
| Less than 5 years | 52% |
| 5 to 10 years | 27.6% |
| 10 to 15 years | 2% |
| 15 to 20 years | 6.1% |
| More than 20 years | 12.2% |
| **Profession field in organisation %** | |
| Civil Engineer | 66.3% |
| Architect | 12.2% |
| Construction Manager | 12% |
| Electrical Engineer | 4.1% |
| Others | 5.1% |

### 4.2. Measurement Model

When analysing reflective measurement models (barriers) in PLS-SEM, it is important to consider their internal consistency, convergent validity, and discriminative validity. After the measurement model's validity and reliability have been confirmed, the structural model will be analysed [91,92]. Table 3 shows that all model constructions that meet the c > 0.70 requirements are valid [93].

**Table 3.** Convergent validity's conclusion.

| Constructs | Average Variance Extracted (AVE) | Composite Reliability | Cronbach's Alpha |
|---|---|---|---|
| Evaluation | 0.509 | 0.861 | 0.808 |
| Preparation | 0.593 | 0.879 | 0.828 |
| Use | 0.575 | 0.843 | 0.749 |

Additionally, as shown in Table 2, all of the buildings were AVE-compliant. AVEs should be more than 0.5 [94], as this is the appropriate quantity. According to the PLS algorithm 3.0, all the estimated AVE values for the components in this study are more than 50% (Table 2). All these numbers show that the measuring model is coherent and consistent within itself. This ensures that each construct (group) is being measured accurately and that the research model is not being used to assess any other construct. High lateral stresses on a structure are indicative of a strong connection between the important parts of any building. Low-outer-loading items (those with a value of less than 0.65 on the scale) must be removed from the weighing process frequently [95]. As can be seen in Figure 2, all first measurement models are approved except for P1, P2, E4, E5, U2, U3, and U4. Therefore, it can withstand any kind of stress from the outside.

Discriminant Validity

Discriminant validity is attained when a concept can be differentiated from other conceptions according to the criteria employed to create the differentiation. Due to this, the construct's original discriminatory validity suggests that it is novel and extends coverage to events that are poorly defined by existing constructs in the model [96]. Some approaches to assess discriminant validity include the Fornell and Larcker [94] criterion, the HTMT (Hetrotrait–Monotrait ratio of correlations), and the criterion.

When evaluating a concept's discriminating validity, it is useful to compare the construct's correlations with all other constructs to the square root of the AVE for that construct. According to Fornell and Larcker [94], the square root of the AVE must be greater than the correlation of the latent variables. The results, as shown in Table 4, support the discriminant validity of the measurement model [97].

**Table 4.** Effective discrimination.

| Constructs | Evaluation | Preparation | Use |
|---|---|---|---|
| Evaluation | 0.714 | | |
| Preparation | 0.525 | 0.77 | |
| Use | 0.573 | 0.674 | 0.758 |

In the present study, the cross-loading criteria were also used by the second method, which established discriminatory validity. This technique makes the counterfactual prediction that an indicator's loading on a given latent construct will be higher than its loading on any other latent construct in any given row. The loading of their construct indicators must be larger than the loading of any rival construct. The assigned latent construct has a greater loading than any cross-loading construct, as seen in Table 5. The outcomes showed that the constructions were very one-dimensional.

**Table 5.** Interactions between variables in measurements.

| Factors | Evaluation | Preparation | Use |
|---|---|---|---|
| E1 | 0.716 | 0.493 | 0.451 |
| E2 | 0.729 | 0.386 | 0.367 |
| E3 | 0.720 | 0.356 | 0.406 |
| E6 | 0.700 | 0.292 | 0.381 |
| E7 | 0.680 | 0.300 | 0.304 |
| E8 | 0.734 | 0.391 | 0.514 |
| P3 | 0.406 | 0.763 | 0.522 |
| P4 | 0.342 | 0.781 | 0.523 |
| P5 | 0.446 | 0.787 | 0.614 |
| P6 | 0.511 | 0.774 | 0.501 |
| P7 | 0.290 | 0.742 | 0.416 |
| U1 | 0.397 | 0.449 | 0.649 |
| U5 | 0.422 | 0.562 | 0.816 |
| U6 | 0.458 | 0.433 | 0.736 |
| U7 | 0.459 | 0.585 | 0.818 |

### 4.3. Path Model Validation

After confirming that the factor for delivering sustainability adoption constitutes a formative construct, we investigate collinearity among the construct's formative objects by calculating the variable inflation factor (VIF). Given that no VIF is over 3.5, each subdomain contributes uniquely to the higher-order structures. The importance of the route coefficients is also predicted using a bootstrapping method. As seen in Figure 3, all of the pathways are significantly different from zero [98].

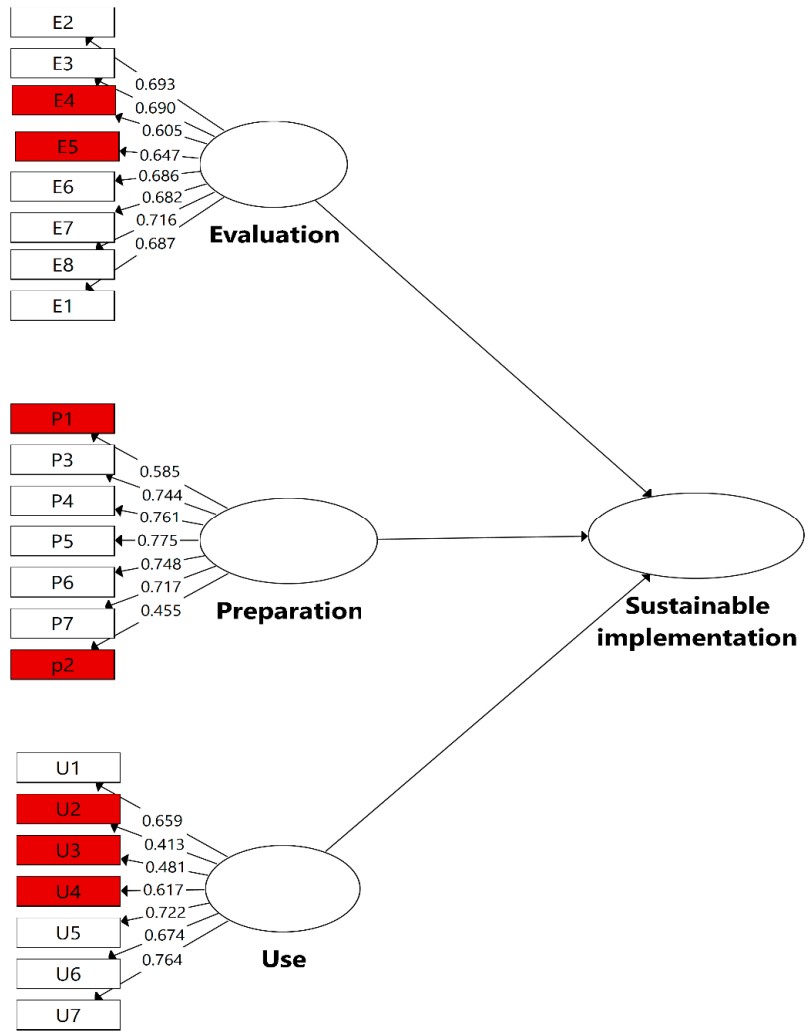

**Figure 2.** Outer loading analysis.

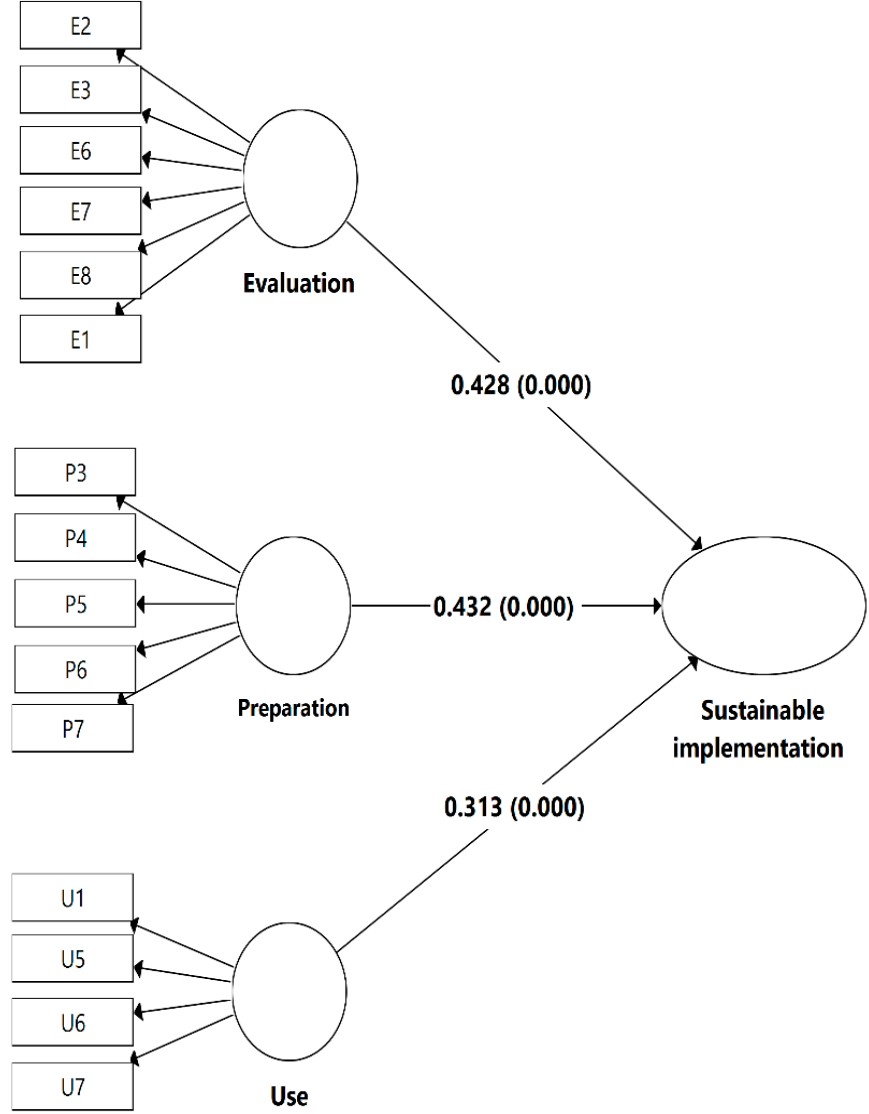

**Figure 3.** The PLS-SEM structural model (β and outer loading values shown on arrows).

## 5. Discussion

Many wealthy countries rely heavily on environmentally responsible building practices, while in poorer countries, this trend is still in its infancy. Egypt, like many other developing nations, has had difficulties and paradoxes in the construction standard. Given this, it's clear that sustainable practices need to be put into place to deal with these problems. The building industry, like the rest of the business, must urgently adopt sustainable practices if we are to reap long-term economic benefits while simultaneously conserving the built environment [8]. Much focus in recent years has been placed on incorporating sustainability into a wide range of fields and methods. However, a survey of existing research reveals a dearth of coverage of construction project management. This is a serious issue because the building sector accounts for close to 40% of world $CO_2$ emissions and 41% of global energy consumption [11]. Regardless of these realities, however, incorporating sustainability into building project management methods has been extremely gradual, with results that are far from satisfying [12]. This reflects the effects of impediments, which have hampered efforts to make sustainable construction project practices the norm. Effective implementation of sustainable parameters will go a long way to society's improved general well-being.

The suggested model shows how the three pillars of sustainability delivery significantly affect the success of sustainability initiatives. This can improve the long-term viability of home construction projects. Therefore, businesses in the construction industry may save money and time without sacrificing quality or functionality by embracing sustainability practices. Next, we will look at how the PLS-SEM model's components may be used to rank the importance of sustainability when it comes to constructing new buildings.

### 5.1. Preparation

Preparation places a premium on the desire and requirement for an invention, as well as knowledge and awareness [99]. It also explains what motivated people to start including environmental considerations in their undertakings [56]. With an outside coefficient of 0.432, the PLS-SEM model recommends that this component has the ultimate influence on the aspects of sustainable delivery through the Preparation component. It should be noted that factors Compliance with anti-corruption rules and regulations in the decision-making process and Enacting required policies in supporting sustainability principles establishment in megaprojects by governmental and professional bodies are ruled out due to a lack of linkage. P5 (knowledge and awareness of sustainable project delivery in the Project Management Team (PMT)) was the factor with the highest level of influence on preparation for the integration of sustainability in megaprojects. Results showed that Pre4 (Effective prerendering and tendering investigations) was the second most influential factor. P3 (Comprehensive contract and specification documentation) and P7 (In-depth research of the contractors' past work to determine their familiarity with the notion of sustainability and their track records in implementing sustainable projects) turned out to have no impact. This finding is consistent with the findings of Hosseini et al. [56]. Understanding and awareness of environmentally responsible methods of project delivery within the Project Management Team (PMT) are very important; in addition, successful tendering and pre-bidding studies should be adopted. Furthermore, the most important aspect of the planning phase was tied explicitly to the level of information and comprehension at hand. In conclusion, the long-term viability of Egypt's construction projects depends on the level of attention paid to environmental concerns.

### 5.2. Evaluation

The second principal component is related to Evaluation. There is a correlation between the strategic goals of the company, the regulatory climate in which the organisation operates, and the backing of key decision-makers when it comes to assessing the value of sustainable measures [56]. Before making any decisions about new technologies, Slaughter [100] stressed the importance of evaluating their potential benefits and drawbacks. By Rogers et al. [101] and Wolfe [102] definitions of innovation dissemination, the steps become clear. To measure the stability of linkages, one may use fit SEM models, since the loadings (correlations) between variables reflect the relative importance of factors in modifying underlying constructs and provide a metric for assessing evaluation factors [103]. It comprises factors, such as establishing a reliable system for strategic planning, stakeholders' firm dedication to the project's long-term success, respect for the interests of parties other than the client, all parties involved have agreed upon and articulated their top priorities, sustainability project outcomes that are in line with stakeholder priorities, and clear goals and boundaries for the project. Compliance with anti-corruption regulations has a significant impact on the review phase of project development. Given the dire circumstances in which developing countries find themselves regarding corruption in megaprojects, this is a logical conclusion [104]. Another key obstacle to sustainable building in Iran is incomplete project scopes and a lack of a systematic strategy for designing megaprojects [105]. There is a complete lack of consideration for current conditions, available resources, or potential strategic or long-term consequences when defining and funding megaprojects in Iran [105]. Unfortunately, political considerations sometimes trump more nuanced strategic considerations when making judgments [106]. A rigorous process for

holding decision-makers responsible for their failings in defining megaprojects is lacking, despite the high failure rate and severe consequences. Symptoms of corruption include a failure to take responsibility for one's actions [107]. This lends credence to the importance placed on combating corruption, as evidenced by a crucial consideration throughout the planning phase. Furthermore, project managers should evaluate the advantages and risks of adopting any innovation, such as GVETs, to justify any investment [108]. They should deploy value management techniques considering all the available alternatives of innovation to select the option with the best feasible value for money [100,109]. The prominence of such evaluation is underpinned further by the cognitive model of technology adoption stating the prominence of viewpoints of practitioners regarding the perceived difficulty and the perceived benefits of adopting a technology [110]. The effects of the outcomes of the activities fulfilled and the decisions made in this stage transcend the initial stages of adopting GVETs and affect the whole performance of GVETs within their whole lifecycle in an organisation [111]. The findings of the study by Kam et al. [112] endorsed such insight by demonstrating that performance records of organisations in different stages of using virtual design and construction are correlated. In addition, alignment with organisational strategies gives rise to organisational support for adopting innovation as the factor deemed essential to facilitate the diffusion of innovation [113]. The effects of demands and needs are highlighted in the literature as well [114]. It is because any organisation should recognise the demands and the need for innovation even before any attempt to acquire information about the innovation [99].

*5.3. Use*

The third principal component is related to ***use***. An improvement in a system or working technique that is novel to the appropriate situation is what is meant by "real usage of an innovation" [56]. This involves factors, such as emphasis on high-quality workmanship, the skill and knowledge of project managers, consistent access to all necessary resources (money, equipment, supplies, etc.) throughout the project's duration, and a transparent and competitive procurement process. Improved construction quality is possible with well-organised building projects. According to Dai et al. [115], construction labour productivity is impacted by a manager's ability to organise, arrange, and lead the job. As a result of an absence of direction and supervision, construction projects often have issues with quality. Since this is the case, it is clear that good construction management is fundamental to any successful building endeavour [116]. Employees who can come up with novel approaches to the problems that the company is now facing are considered to be "idea generators" [117]. To that end, gatekeepers actively seek out and analyse novel options as they emerge [118]. They are crucial in the building industry when it comes to spreading new ideas [100,119]. This means that idea promotion and generation are prerequisites for every action or technique that results in the adoption and spread of innovation in any setting [120], including the building industry [121]. Verburg et al. [122] implication that the dissemination of knowledge of known good practices is vital in encouraging acceptance of any innovation, including GVETs, in construction projects is supported by this finding.

*5.4. Managerial Consequences*

Rearranging the factors for sustainable adoption can help produce a roadmap that stakeholders such as project owners and contractors can follow to overcome the barriers and embrace sustainability in the construction sector. Additionally, a standard for a practical framework for the effective transformation of construction participants through sustainable phases and activities may be established due to this reorganisation. The research results will help Egypt get closer to its goal of creating a prosperous, environmentally sustainable economy that can compete successfully in global markets. This research's findings can also inspire the implementation of sustainable practices in construction projects in other developing countries [123]. Since developing nations have many more obstacles to overcome, such as paying for expensive environmental solutions, this is especially

important there [124]. These nations may have the chance to incorporate performance into the design methods of construction projects if they follow sustainable practices [125,126]. However, this research makes a substantial contribution that has significant consequences for the construction sector in the following ways:

- It provides a database of connected aspects with sustainability delivery factors to help businesses determine how to remain competitive and successful in a global market.
- It helps owners, consultants, and contractors evaluate and decide on sustainable practices to improve construction projects' consistency, efficiency, and effectiveness.
- It provides actual facts that might ease the path to sustainability adoption in Egypt and other developing countries.
- The United Kingdom, the United States, Hong Kong, Australia, and other countries, including Malaysia, China, and Saudi Arabia, have been the primary foci of sustainability and sustainability research in the building industry. As a result, there is a dearth of literature on sustainability in developing countries and no studies focusing on its application in the Egyptian building sector. Consequently, our study has effectively established a bridge between sustainability and the Egyptian building sector. This paves the way for a robust conversation on sustainability as a tool for enhancing the safety of regional construction projects and ending a knowledge gap.
- The results of this study can help improve the sustainability of future construction in Egypt. Our research explains why sustainability initiatives are implemented to reduce wasteful spending and ensure that resources are allocated fairly amongst different projects. This way, everyone involved in the project can concentrate on its budget, schedule, and efficacy to achieve its goals. Achieving a high level of success in a project has a beneficial effect in the long run.
- The findings of this study may also be used as a standard by which future projects can be measured, as well as a roadmap for minimising the difficulties inherent in their implementation. Things such as budget overruns, finishing projects on time, and vague requirements all made a list. In addition, business owners and managers may use this study's findings to understand better how incorporating sustainable practices might contribute to the success of their initiatives.

### 5.5. Implications for Theory

Although the idea of developing sustainable concepts is not novel [127], it looks to be playing an increasingly significant role in many companies [128]. However, there appears to be a lack of research on the elements that lead to the adoption of sustainable practices in Egypt's construction industry. As a first step, this research uses empirical methods to pinpoint the main factors of sustainability, which may be used to understand better how to introduce these principles into the building sector. Researchers, especially those in construction management, might use this data to investigate the sustainability challenges in third-world nations. This analysis's theoretical components provide a quantitative basis for locating the sustainability hurdles, which may be put to good use in Egypt and other developing countries.

### 6. Conclusions

Sustainable building principles should be included at every step of the planning process for maximum benefit without compromising the structure's intended function. This study suggests a mathematical model of the main parameters influencing the timely and cost-effective completion of green construction projects. The study is based on responses from 95 construction industry professionals in Egypt. Information was combined into a structural equation model to produce a predictive model. The present study contributes to the body of knowledge in three significant ways. First, the situation in Egypt, a significant emerging country with megaprojects, is underrepresented in the literature. Second, prior studies have not adequately incorporated contextual aspects into their examinations of sustainability problems, even though sustainability is highly contextual. Finally, Egypt's

situation represents the broader trend among developing countries to prioritise economic growth above sustainability, as Egypt is an emerging country with enormous resources but severe developmental issues. Egypt is a case study for the issue of sustainability, and the lessons it can teach the rest of the developing world are universal. Egypt is a primary emerging market making substantial investments in construction methods and infrastructure. This research aims to shed light on that critical issue. This study is among the earliest academic investigations into this topic and sheds new light on the aspects that determine the long-term viability of construction endeavours in Egypt. Another contribution is the provision of a structural equation modelling (SEM) model that evaluates the influence of various factors on sustainability. According to the findings of the PLS-SEM analysis, the variables connected to preparation are the essential elements that affect substantial delivery. As a result of this research, the authorities in Egypt's building sector will have a road map for implementing sustainability principles to reduce building costs, boost the local ecosystem, and strengthen social cohesion. These goals were established considering the research that was conducted.

### 7. Limitations and Future Directions

It is important to note that there are certain caveats to the present study's findings, despite its merits. The first is that the sample size is small, calling for additional confirmation of the SEM model estimates and conclusions with more extensive samples. In the second step, the researchers decided which variables to incorporate into the model. Expert opinion is widely accepted as a valid research method, although its reliance on subjective evaluation leaves it vulnerable to criticism. There must be more studies on this, ideally with a wider variety of sustainability criteria and other points of view. In addition, construction professionals and legislators must examine and accept the presented ideas in terms of practicality and efficacy. This calls for more study to hone the proposals and create frameworks and procedures for implementing sustainability in megaprojects throughout Egypt's building sector. One potentially fruitful area for investigation would be to use the study's findings as a basis for repeating the experiment in different settings. Examining how the findings may be adapted for use in other developing nations is of particular interest.

**Author Contributions:** Research idea, A.F.K.; conceptualization, A.F.K. and M.M.H.; methodology, A.F.K.; software, A.F.K.; validation, M.M.H. and A.F.K.; formal analysis, A.F.K.; investigation, A.F.K.; resources, M.M.H. and A.F.K.; data curation, M.M.H. and A.F.K.; writing—original draft preparation, A.F.K. and M.M.H.; writing—review and editing, A.F.K. and M.M.H.; visualization, A.F.K. and M.M.H.; supervision, A.F.K.; project administration, A.F.K.; funding acquisition, A.F.K. and M.M.H. All authors have read and agreed to the published version of the manuscript.

**Funding:** This research received no external funding.

**Institutional Review Board Statement:** Not applicable.

**Informed Consent Statement:** Not applicable.

**Data Availability Statement:** Not applicable.

**Conflicts of Interest:** The authors declare no conflict of interest.

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
