# Peer review of "Exploring the Sustainable Delivery of Building Projects in Developing Countries: A PLS-SEM Approach"

_sustainability, doi:10.3390/su142215460_

Round 1
Reviewer 1 Report
Do not use pronouns in the paper.
The abstract is not informative and its scope and methodology are not well structured and not clear.
The question/gap that the paper will address are not clear in the introduction.
"Incorporating Sustainability into the Construction Industry" and "Factors affecting the Sustainable Delivery of building projects". Are these section present the literature review. If the answer is yes, the gaps in the prior works have not been illustrated. In Egypt, there are several studies on the sustainability. Further, the factors of Table 1 are very low and they did not reflect all the pillars of the sustainability.
Methods of Research and Model Construction. This section must be restructured to inform on a step-by-step basis the phases of the methodology.
The content, piloting, validity, and reliability of the questionnaire or the response are not mentioned.
The discussion section need more analysis to describe the findings of the paper along with the prior relevant works.
Author Response
Reviewer 1
Do not use pronouns in the paper.
Many thanks for the reviewer’s comment. The paper has been enhanced and pronouns have been avoided
The abstract is not informative and its scope and methodology are not well structured and not clear.
The abstract has been enhanced. Kindly refer to abstract lines 11-21:
“ Abstract: Sustainable building approaches should be included at every level of the development process for maximum benefit without jeopardizing the structure's intended usage. The purpose of this study is to investigate the numerous determinants of Sustainable delivery in the construction industries of developing nations. Existing literature was used to inform the development of a closed-ended questionnaire, which was then administered to relevant construction experts after being contextualized. Consequently, this research used Partial Least Squares Structural Equation Modelling (PLS-SEM) to assess the interrelationships among the various classifications we created for these variables. According to the PLS-SEM analysis, the factors most strongly influencing sub-stantiality delivery are those associated with preparation. As a result of this research, authorities in Egypt's building sector will have a road map for implementing sustainability principles to reduce building costs, boost the local eco-system, and strengthen social cohesion.”
The question/gap that the paper will address are not clear in the introduction.
"Incorporating Sustainability into the Construction Industry" and "Factors affecting the Sustainable Delivery of building projects". Are these section present the literature review. If the answer is yes, the gaps in the prior works have not been illustrated. In Egypt, there are several studies on the sustainability. Further, the factors of Table 1 are very low and they did not reflect all the pillars of the sustainability.
We are grateful for the reviewer’s comments. The gap has been highlighted based on the discussion on the previous studies. Kindly refer to lines 109-156:
“For a construction project to be sustainable, it must serve many, often conflicting goals, including environmental protection, social betterment, and the advancement of the construction company's strategic interests and financial well-being [41]. What this implies is that sustainable practises will be included into every stage of the project's development and implementation [42, 43]. Keeping this in mind, the current body of research on sustainability may be broken down into a few distinct groups. Standards [44], sustainable resources [43], eco-design, and corporate responsibility are only a few of the topics that have been researched in the realm of construction [41]. Stakeholder management [45], lifecycle management [46], and sustainability assessment are all examples of related topics that arise throughout the delivery phase of a project [47, 48]. Sustainable project organisations [49], sustainable project practises [14], and sustaina-ble project management [49] have also been the subject of research. Those developing nations with a penchant for sustainability have often adopted Western techniques [14, 50, 51].
The triple bottom line (TBL) is a widely used framework for understanding sus-tainable development. It considers not just economic but also social and environmental (or ecological) considerations (or financial). People, planet, and profit are sometimes referred to as the “three Ps”. These three elements, as defined by Willard [52], make up the foundation of sustainable development. Organizations in the construction sector have been driven to adopt the concept of sustainability on building projects as a result of the growing trend toward clean production, green products, and an increased awareness of climate change [42, 53]. Social and environmental advancement, on the one hand, and the construction company's strategic interests and bottom line, on the other, are often at odds with one another, making it difficult to achieve sustainability in building projects [41]. Taking this approach entails incorporating sustainability ide-as into the project's actions from the beginning to the end [42, 43]. Keeping this in mind, there are numerous groups into which the existing body of sustainability re-search can be sorted [12]. Standardization research [44], sustainable materials research [43], eco-design research, and CSR research are all examples from the building sector [41]. Stakeholder management [45], lifecycle management [46], and sustainability as-sessment are three examples of such groups involved in the actual delivery of projects [48, 54]. There have also been studies into sustainable project management [49], sus-tainable project organisations [55], sustainable project practises [14], and sustainable decision-making [55]. Countries in the developing world that are making efforts to-ward sustainability often model their policies and procedures after those of more in-dustrialised nations [14, 50, 51]. However, the criteria that determine the sustainability of construction projects are extremely context specific, and the methods used in ad-vanced nations may not be applicable in less developed nations [14]. To far, there has been less academic inquiry into how sustainability principles could interface with building initiatives in Egypt. Most of the ones done so far have taken an exploratory approach, focusing on certain subfields of sustainability. Though limited, the results do indicate delivery issues in underdeveloped nations [56]. Environmentally friendly, re-source-efficient "sustainable buildings" have been called for in the literature [19]. In addition, by 2030, the Egyptian government hopes to have Egypt ranked among the top 30 countries in the world [57]. As a result, sustainable practises should be incorpo-rated into Egyptian construction projects [58]. It's important to note that the factors that affect the long-term viability of construction projects vary greatly depending on their location; as a result, practises common in more developed countries may not be appropriate in less economically stable regions [14]. Table 1 presents the most relevant parameters that determine sustainable delivery for building projects, based on the study of Hosseini, et al. [56].”
In addition, the research question has been raised and highlighted in the introduction section. Kindly refer to lines 68-69:
” What factors affect the efficiency and affordability of building projects in developing nations?”
Methods of Research and Model Construction. This section must be restructured to inform on a step-by-step basis the phases of the methodology.
We fully concur with the reviewer. The methods section has been restructured. Kindly refer to lines 158-219:
“3. Methods of Research and Model Construction
The drive of this study is to catalogue and evaluate Sustainable Building Project Delivery in Egypt. Following a thorough analysis of the available literature, 22 contributors to long-term implementation were recognized, as demonstrated in Table 1. The questionnaire poll was conducted by sending out a list of obstacles to home construction industry professionals. This was done so that the aspects to implementing sustainability would be as clear and comprehensive as possible. For this reason, the methodology followed Figure 1. Adopted from Kineber, et al. [58].
3.1 Literature review
Different steps of data collecting, refinement, and categorization have been used, and the literature review technique has been investigated through (exploratory) research [70]. Articles, research papers, government documents, green construction codes, and other materials have all been carefully considered to compile the data for this study. Information is whittled down in the review process by being chosen, simplified, and abstracted. This refers to organizing information into broad concepts [71]. Hosseini, et al. [56] categorization was used as a basis for this classification, and care was taken to assign relevant material to the appropriate subgroups (such as subthemes) within the main categories [70]. We broke down the whole thing into global, regional, and country categories, with a focus on developing nations. Plus, the most important concerns raised by construction projects.
3.2 Questionnaire survey
A larger range of possible participants in Egypt's residential building sector was recruited for a questionnaire study to learn more about the barriers to sustainability implementation in Egypt. The demographics of the respondents, the barriers to implementing sustainability, and the perspectives of respondents themselves made up the three parts of this study (Table 1). The questionnaire was made to allow for free-form responses (to add any barriers that the participants considered essential). Those involved in the process were notified, including contractors, consultants, and customers. These professionals include architects, electrical engineers, quantity surveyors, structural and mechanical engineers. Based on their own understanding and observations, respondents assigned numerical values to these aspects: 5 for very high, 4 for high, 3 for normal, 2 for small, and 1 for nil or very little. Previous research has used this scale[58, 72-76]. Considering the novelty of the concept of sustainability in Egypt, stratified sampling was used to investigate its effects on a subset of the population. In addition, the sample size used in this study was decided by a purpose analysis of the methodology [77]. 95 out of 150 people were contacted for this study, with a response rate of around 70%, in accordance with Yin [78] recommendation that a small sample size is ideal for doing PLS-SEM. According to the findings of these studies, this rate of return is satisfactory [79, 80].
3.3 Analytical approach (PLS-SEM)
To better understand what goes into making a building project sustainable, four models were reviewed from the literature and compared to the best option made possible by sustainable delivery. Multiple Linear Regressions, Structural Equation Modeling, System Dynamic, and Artificial Neural Networks are the methods considered (ANN). This is because there is a connection between the variables that has prevented us from using the regression equation. In a major way, this constrains the applicability of the regression equation [81]. System dynamics could not be used because the data utilized in the study had no temporal context. The study's fundamental purpose is to investigate what elements contribute to long-term sustainability in delivery using an artificial neural network as a forecasting tool. To meet the needs of the research, the SEM technique describes the relationship between as many observable and unobservable variables as are necessary. According to Amaratunga, et al. [82], SEM is an effective method for addressing the presence of error in variables. The study used the structural equation modelling technique to create a model and determine the connection between elements related to sustainable delivery. To fill the gap left by the lack of familiarity with hypothesis testing procedures, Byrne [83] argued that structural equation modelling (SEM) has become a standard non-experimental research tool. Also, using research articles from the MIS Quarterly, Ringle, et al. [84] concluded that this technique has improved with time. Yuan, et al. [85] also found that SEM is a common and widely used method for analyzing social data. The SEM method was utilized in the study since it is widely known to be effective in the construction industry [86]. Additionally, the activities in the sustainable delivery elements have been evaluated using a PLS model that includes both reflective and formative variables.”
The content, piloting, validity, and reliability of the questionnaire or the response are not mentioned.
Many thanks for the constructive comments. We agree with the reviewer. The above-mentioned analyses have been explored and added. Kindly refer to lines 223-266:
“Analysis and Findings
4.1 Pilot survey
After consulting with executives, market researchers, and academics, the initial set of questionnaire questions was drafted. In order to guarantee the reliability of po-tential data gathering techniques, a Pilot test method was incorporated on a modest scale [87]. A pilot study is conducted to test the research design methodologies and see if the procedures and methods used in the full study are adequate [87]. Workmates, experts, and superiors all provided feedback that led to revisions to the questions. In most cases, questions on the questionnaire were reworded as part of the instrument's revisions.
4.2 Reliability analysis
Quality and efficiency in analysis are of utmost importance, so it's important to have a well-thought-out theory framework, as well as acceptable protocols and meth-odologies, before beginning any investigation. Research findings are trusted because of reliability and validity, as stated by Neuman [88]. Testing for reliability is often per-formed to rule out biases and flaws in the analysis and guarantee that the same results will be achieved if the researcher repeats the same technique. This contrasts with va-lidity, which is based on objective evaluations of the study's quality and approxima-tion to the truth. The method of data collecting was determined after reliability testing to guarantee repeatable results. However, face validity and content validity are the two conventional measures of validity used to evaluate whether the components of the survey instrument are appropriate and reflective of the intended research design [89]. To ensure the reliability of the research instrument, a sample of 12 research profes-sionals from the field and the academy were chosen at random. This was done in ac-cordance with the suggestion made by Sushil and Verma [90], that face validity is checked by having experts assess the contents to test them and ensure they appear suf-ficient. Experts analyze the substance of the test to ensure the items seem adequate (face validity). To determine the validity and reliability of the findings, Cronbach's al-pha was utilized. The reliability of the scale was assessed by examining the correlations among the items in the sample to determine whether they were indeed related. Values of Cronbach's alpha above.8 are considered very reliable, while values between.6 and.7 are considered adequate. Since each of the alpha Cronbach values is more than that, this means that they are all suitable. Six, the average set correlation was greater than 0.3, suggesting that there were stable internal variables present in all objects [91].
4.3 Demographic analysis
6.1 Respondents' characteristics and demographic profiles
As shown in Table 2, this study looked at participants' comprehension and awareness of the sustainable construction. According to the results, respondents were "acquainted" and "completely familiar" with 46.9% and 24.5% of the sample, respec-tively. The results also revealed that "Professionality" emphasized that "Architect" (12.2.7%) and "Civil Engineer" (66.3%) were seen to have the highest frequency and "Electrical Engineer" (4.4%) to have the lowest frequency, respectively. For bachelor's, master's, and doctoral degrees, the qualifications of the respondents were measured at (65.3%, 13.3%, and 18.4%, respectively). Table 2 also reveals that about 52% of re-spondents had a job for between one year and five years or less. Approximately (27.6%, 2%), and (6.1%) of respondents had job experience ranging from 5 to 10 years, 11 to 15 years, and more than 25 years, respectively. This suggests that the study's participants are knowledgeable and able to learn from it.”
The discussion section needs more analysis to describe the findings of the paper along with the prior relevant works.
We are grateful for the reviewer’s comments. Discussion and findings have been updated based on the previous studies. Kindly refer to lines 318-429:
“Many wealthy countries rely heavily on environmentally responsible building practices, while in poorer countries, this trend is still in its infancy. Egypt, like many other developing nations, has had difficulties and paradoxes in the construction standard. Given this, it's clear that sustainable practices need to be put into place to deal with these problems. The building industry, like the rest of business, must urgently adopt sustainable practices if we are to reap long-term economic benefits while simultaneously conserving the built environment [8]. Much focus in recent years has been placed on incorporating sustainability into a wide range of fields and methods. However, a survey of existing research reveals a dearth of coverage of construction project management. This is a serious issue because the building sector accounts for close to 40% of world CO2 emissions and 41% of global energy consumption [11]. Regardless of these realities, however, incorporating sustainability into building project management methods has been extremely gradual, with results that are far from satisfying [12]. This reflects the effects of impediments, which have hampered efforts to make sustainable construction project practices the norm. Effective implementation of sustainable parameters will go a long way to society's improved general well-being.
The suggested model shows how the three pillars of sustainability delivery significantly affect the success of sustainability initiatives. This can improve the long-term viability of home construction projects. Therefore, businesses in the construction industry may save money and time without sacrificing quality or functionality by embracing sustainability practices. Next, we'll look at how the PLS-SEM model's components may be used to rank the importance of sustainability when it comes to constructing new buildings.
6.1 Preparation
Preparation places a premium on desire and requirement for an invention, as well as knowledge and awareness [100]. It also explains what motivated people to start including environmental considerations in their undertakings [56]. With an outside coefficient of 0.432, the PLS-SEM model recommends that this component has the ultimate influence on the aspects of sustainable delivery through the Preparation component. It should be noted that factors Compliance with anti-corruption rules and regulations in the decision-making process and Enacting required policies in supporting sustainability principles establishment in megaprojects by governmental and professional bodies are ruled out due to a lack of linkage. P5 [knowledge and awareness of sustainable project delivery in the Project Management Team (PMT)] was the factor with the highest level of influence on preparation for integration of sustainability in megaprojects. Results showed that Pre4 (Effective prerendering and tendering investigations) was the second most influential factor. P3 (Comprehensive contract and specification documentation) and P7 (In-depth research of the contractors' past work to determine their familiarity with the notion of sustainability and their track records in implementing sustainable projects) turned out to have no impact. This finding is consistent with the findings of Hosseini, et al. [56]. That understanding and awareness of environmentally responsible methods of project delivery within the Project Management Team (PMT) is very important in addition, successful tendering and pre-bidding study should be adopted. Furthermore, the most important aspect of the planning phase was tied explicitly to the level of information and comprehension at hand. In conclusion, the long-term viability of Egypt's construction projects depends on the level of attention paid to environmental concerns.
6.2 Evaluation
The second principal component is related to Evaluation. There is a correlation between the strategic goals of the company, the regulatory climate in which the organization operates, and the backing of key decision makers when it comes to assessing the value of sustainable measures [56]. Before making any decisions about new technologies, Slaughter [101] stressed the importance of evaluating their potential benefits and drawbacks. In accordance with Rogers, et al. [102] and Wolfe [103] definitions of innovation dissemination, the steps become clear. To measure the stability of linkages, one may use fit SEM models, since the loadings (correlations) between variables reflect the relative importance of factors in modifying underlying constructs and provide a metric for assessing evaluation factors [104]. It comprises factors, such as establishing a reliable system for strategic planning, stakeholders' firm dedication to the project's long-term success, respect for the interests of parties other than the client, all parties involved have agreed upon and articulated their top priorities, sustainable project outcomes that are in line with stakeholder priorities and clear goals and boundaries for the project. Compliance with anticorruption regulations has a significant impact on the review phase of project development. Given the dire circumstances in which developing countries find themselves regarding corruption on megaprojects, this is a logical conclusion [105]. Another key obstacle to sustainable building in Iran is incomplete project scopes and a lack of a systematic strategy to designing megaprojects [106]. There is a complete lack of consideration for current conditions, available resources, or potential strategic or long-term consequences when defining and funding megaprojects in Iran [106]. Unfortunately, political considerations sometimes trump more nuanced strategic considerations when making judgments [107]. A rigorous process for holding decision makers responsible for their failings in defining megaprojects is lacking, despite the high failure rate and severe consequences. Symptoms of corruption include a failure to take responsibility for one's actions [108]. This lends credence to the importance placed on combating corruption, as evidenced by a crucial consideration throughout the planning phase. Furthermore, Project managers should evaluate the advantages and risks of adopting any innovation, such as GVETs, to justify any investment [109]. They should deploy value management techniques considering all the available alternatives of an innovation to select the option with the best feasible value for money [101, 110]. The prominence of such evaluation is underpinned further by the cognitive model of technology adoption stating the prominence of viewpoints of practitioners regarding the perceived difficulty and the perceived benefits of adopting a technology [111]. The effects of the outcomes of the activities fulfilled and the decisions made in this stage transcend the initial stages of adopting GVETs and affects the whole performance of GVETs within their whole lifecycle in an organization [112]. The findings of the study by Kam, et al. [113] endorsed such an insight by demonstrating that performance records of organizations in different stages of using virtual design and construction are correlated. In addition, alignment with organizational strategies gives rise to organizational support for adopting the innovation as the factor deemed essential to facilitate diffusion of an innovation [114]. The effects of demands and needs are highlighted in the literature as well [115]. It is because any organization should recognize the demands and the needs for an innovation even prior to any attempt to acquire information about the innovation [100].
6.3 Use
The third principal component is related to Use. An improvement in a system or working technique that is novel to the appropriate situation is what is meant by "real usage of an innovation" [56]. This involves factors, such as emphasis on high-quality workmanship, the skill and knowledge of project managers, consistent access to all necessary resources (money, equipment, supplies, etc.) throughout the project's duration, transparent and competitive procurement process. Improved construction quality is possible with well-organized building projects. According to Dai, et al. [116], construction labor productivity is impacted by a manager's ability to organize, arrange, and lead the job. As a result of a absence of direction and supervision, construction projects often have issues with quality. Since this is the case, it's clear that good construction management is fundamental to any successful building endeavor [117]. Employees who are able to come up with novel approaches to the problems that the company is now facing are considered to be "idea generators" [118]. To that end, gatekeepers actively seek out and analyze novel options as they emerge [119]. They are crucial in the building industry when it comes to spreading new ideas [101, 120]. This means that idea promotion and generation are prerequisites for every action or technique that results in the adoption and spread of an innovation in any setting [121], including the building industry [122]. Verburg, et al. [123] implication's that the dissemination of knowledge of known good practices is vital in encouraging acceptance of any innovation, including GVETs, in construction projects is supported by this finding.”

Reviewer 2 Report
①Figures 2 and 3 need to be optimized.
②There are too many quotations. It is recommended to simplify the quoted articles.
③Directly using the standards of other articles, there is no reasonable explanation and explanation of the feasibility of influencing factors.
④Please explain the specific meaning of "Evaluation", "Preparation" and "Use".
Author Response
Reviewer 2
①Figures 2 and 3 need to be optimized.
Many thanks for the reviewer valuable comment. The figures have been enhanced and optimized. Kindly refer to Figure 2 and 3
②There are too many quotations. It is recommended to simplify the quoted articles.
Many thanks for the reviewer valuable comment. The unwanted quotations have been removed
③Directly using the standards of other articles, there is no reasonable explanation and explanation of the feasibility of influencing factors.
Many thanks for the reviewer comment. The logical justification and justification of the viability of influencing variables have been added. Kindly refer to section 6. Discussion.
④Please explain the specific meaning of "Evaluation", "Preparation" and "Use".
Many thanks for the reviewer comment. We agree with the reviewer the full meaning for the above mentioned constructs has been added. Kindly refer to discussion part

Round 2
Reviewer 1 Report
All of all, the references were 98. Now, they become 129 references without any need. Most of the added references are not associated with the sustainability, which is the scope of the study. Further, the authors are not accurate when they describe their work.
The abstract still is in-informative.
The authors listed in the abstract that they will the evaluate the interrelationships between the groups of the variables. The interrelationship means the effect of factor on another one. These effects are not appear in the study.
The question of the study is "What factors affect the efficiency and affordability of building projects in developing nations?". However, the outcomes of the models are use, preparation, and evaluation.
I read the methodology and the analysis and findings several times and I struggle to understand its sequences. It is the first time to find that the pilot survey is among the analysis and findings section. As I know, the methodology must be as following: a) defining the factors, b) Inserting the factors in questionnaire, c) piloting the questionnaire, d) determining the population and sampling method, e) checking the validity and reliability of the responses, f) checking the adequacy of the responses. Then, illustrating the analytical techniques. However, in this paper, the methodology are not arranged. For instance, I can not understand the purpose of these sentences " To better understand what goes into making a building project sustainable, four models were reviewed from the literature and compared to the best option made possible by sustainable delivery. Multiple Linear Regressions, Structural Equation Modeling, System Dynamic, and Artificial Neural Networks are the methods considered (ANN). This is because there is a connection between the variables that has prevented us from using the regression equation. In a major way, this constrains the applicability of the regression equation [81]. System dynamics could not be used because the data utilized in the study had no temporal context". What is the purpose of reviewing of these models. Are the ANN is used in the study. The authors are not accurate when they describe their work. The methodology is the core of the research. Thus, it needs to be more clarified and arranged better with a major focus on the study. This is a critical point and needs to be considered.
The language of the paper needs more improvements.
Author Response
All of all, the references were 98. Now, they become 129 references without any need. Most of the added references are not associated with the sustainability, which is the scope of the study. Further, the authors are not accurate when they describe their work.
Thank you very much for reviewing our manuscript. We also greatly appreciate the will complimentary comments and suggestions. We have carried out the modifications that the reviewer suggested and revised the manuscript accordingly. Consequently, the meaning of whole paper has been enhanced to reflect the scope of the paper. In addition, the increasing number of references is based on the comments received from the second reviewer
The abstract still is in-informative.
Many thanks for the reviewer’s valuable comment. We agree with the reviewer, accordingly, the abstract has been enhanced based on the reviewer’s valuable comments. Kindly refer to abstract lines 11-24:
“Sustainable building approaches should be included at every level of the development process for maximum benefit without jeopardizing the structure's intended usage. However, researchers have paid less attention to how they may be applied to developing nations. The purpose of this study is to investigate the numerous determinants of Sustainable delivery in the construction industries of developing nations. For this to happen, existing literature was used to inform the development of a closed-ended questionnaire. Consequently, the importance of these factors investigated by 95 structured questionnaires by building professionals. As a results, the factors structure was determined and confirmed using the study's partial least square structural equation modeling (PLS-SEM) approach, which was also utilized to investigate the connections between these factors. According to the PLS-SEM analysis, the factors most strongly influencing substantiality delivery are those associated with preparation followed by evaluation and use factors respectively. As a result of this research, authorities in Egypt's building sector will have a road map for implementing sustainability principles to reduce building costs, boost the local ecosystem, and strengthen social cohesion.”
The authors listed in the abstract that they will the evaluate the interrelationships between the groups of the variables. The interrelationship means the effect of factor on another one. These effects are not appeared in the study.
Many thanks for the reviewer valuable comments. We revised the meaning accordingly and mentioned the relevant meaning on the abstract. In addition, the impact here is refer to impact of each factor on other factor on the same construct. Accordingly, we clearly indicate this impact on the paper. Kindly refer to line 272-311:
“4 Measurement model
When analyzing reflective measurement models (barriers) in PLS-SEM, it is important to consider their internal consistency, convergent validity, and discriminative validity. After the measurement model's validity and reliability have been confirmed, the structural model will be analyzed [92, 93]. Table 2 shows that all model constructions that meet the c > 0.70 requirements are valid [94].
Table 3. Convergent validity's conclusion
Constructs |
Average Variance Extracted (AVE) |
Composite Reliability |
Cronbach's Alpha |
Evaluation |
0.509 |
0.861 |
0.808 |
Preparation |
0.593 |
0.879 |
0.828 |
Use |
0.575 |
0.843 |
0.749 |
Also, as shown in Table 2, all of the buildings were AVE-compliant. AVEs should be more than 0.5 [95], as this is the appropriate quantity. According to the PLS algorithm 3.0, all the estimated AVE values for the components in this study are more than 50% (Table 2). All these numbers show that the measuring model is coherent and consistent within itself. This ensures that each construct (group) is being measured accurately and that the research model is not being used to assess any other construct. High lateral stresses on a structure are indicative of a strong connection between the important parts of any building. Low-outer-loading items (those with a value of less than 0.65 on the scale) must be removed from the weighing process frequently [96]. As can be seen in Figure 2, all first measurement models are approved except for P1, P2, E4, E5, U2, U3, and U4. Therefore, it can withstand any kind of stress from the outside.
4.4.1 Discriminant validity
Discriminant validity is attained when a concept can be differentiated from other conceptions according to the criteria employed to create the differentiation. Because of this, the construct's original discriminatory validity suggests that it is novel and extends coverage to events that are poorly defined by existing constructs in the model [97]. Some approaches to assess discriminant validity include the Fornell and Larcker [95] criterion, the HTMT (Hetrotrait-Monotrait ratio of correlations), and the criterion.
When evaluating a concept's discriminating validity, it is useful to compare the construct's correlations with all other constructs to the square root of the AVE for that construct. According to Fornell and Larcker [95], the square root of the AVE must be greater than the correlation of the latent variables. The results, as shown in Table 3, support the discriminant validity of the measurement model [98].
Table 4: Effective discrimination
Constructs |
Evaluation |
Preparation |
Use |
Evaluation |
0.714 |
||
Preparation |
0.525 |
0.77 |
|
Use |
0.573 |
0.674 |
0.758 |
In the present study, the cross-loading criteria were also used by the second method, which established discriminatory validity. This technique makes the counterfactual prediction that an indicator's loading on a given latent construct will be higher than its loading on any other latent construct in any given row. The loading of their construct indicators must be larger than the loading of any rival construct. The assigned latent construct has a greater loading than any cross-loading construct, as seen in Table 4. The outcomes showed that the constructions were very one-dimensional.
Table 5. Interactions between variables in measurements
Factors |
Evaluation |
Preparation |
Use |
E1 |
0.716 |
0.493 |
0.451 |
E2 |
0.729 |
0.386 |
0.367 |
E3 |
0.720 |
0.356 |
0.406 |
E6 |
0.700 |
0.292 |
0.381 |
E7 |
0.680 |
0.300 |
0.304 |
E8 |
0.734 |
0.391 |
0.514 |
P3 |
0.406 |
0.763 |
0.522 |
P4 |
0.342 |
0.781 |
0.523 |
P5 |
0.446 |
0.787 |
0.614 |
P6 |
0.511 |
0.774 |
0.501 |
P7 |
0.290 |
0.742 |
0.416 |
U1 |
0.397 |
0.449 |
0.649 |
U5 |
0.422 |
0.562 |
0.816 |
U6 |
0.458 |
0.433 |
0.736 |
U7 |
0.459 |
0.585 |
0.818 |
“
The question of the study is "What factors affect the efficiency and affordability of building projects in developing nations?". However, the outcomes of the models are use, preparation, and evaluation.
Many thanks for the reviewer comment. Yes, these outputs are the constructs which inside it have an indicators (these indicators represent the most significate factors). Consequently, This paper is the first attempt to draw attention to these elements.
I read the methodology and the analysis and findings several times and I struggle to understand its sequences. It is the first time to find that the pilot survey is among the analysis and findings section. As I know, the methodology must be as following: a) defining the factors, b) Inserting the factors in questionnaire, c) piloting the questionnaire, d) determining the population and sampling method, e) checking the validity and reliability of the responses, f) checking the adequacy of the responses. Then, illustrating the analytical techniques.
Many thanks for the reviewer valuable comment and we are grateful to the reviewer for the constructive comment. Consequently, the methods part has been initiated and modified based on the reviewer comment. Kindly refer to lines 163-251:
“Methods of Research and Model Construction
The drive of this study is to catalogue and evaluate Sustainable Building Project Delivery in Egypt. Following a thorough analysis of the available literature, 22 contributors to long-term implementation were recognized, as demonstrated in Table 1. The questionnaire poll was conducted by sending out a list of obstacles to home construction industry professionals. This was done so that the aspects of implementing sustainability would be as clear and comprehensive as possible. For this reason, the methodology is followed in Figure 1. Adopted from Kineber, et al. [58].
3.1 Defining the factors
Different steps of data collecting, refinement, and categorization have been used, and the literature review technique has been investigated through (exploratory) research [70]. Articles, research papers, government documents, green construction codes, and other materials have all been carefully considered to compile the data for this study. Information is whittled down in the review process by being chosen, simplified, and abstracted. This refers to organizing information into broad concepts [71]. Hosseini, et al. [56] categorization was used as a basis for this classification, and care was taken to assign relevant material to the appropriate subgroups (such as subthemes) within the main categories [70]. We broke down the whole thing into global, regional, and country categories, with a focus on developing nations. Plus, the most important concerns are raised by construction projects.
3.2 Pilot survey
After consulting with executives, market researchers, and academics, the initial set of questionnaire questions was drafted. To guarantee the reliability of potential data-gathering techniques, a Pilot test method was incorporated on a modest scale [72]. A pilot study is conducted to test the research design methodologies and see if the procedures and methods used in the full study are adequate [72]. Workmates, experts, and superiors all provided feedback that led to revisions to the questions. In most cases, questions on the questionnaire were reworded as part of the instrument's revisions.
3.2 Questionnaire survey
A larger range of possible participants in Egypt's residential building sector was recruited for a questionnaire study to learn more about the barriers to sustainability implementation in Egypt. The demographics of the respondents, the barriers to implementing sustainability, and the perspectives of the respondents themselves made up the three parts of this study (Table 1). The questionnaire was made to allow for free-form responses (to add any barriers that the participants considered essential). Those involved in the process were notified, including contractors, consultants, and customers. These professionals include architects, electrical engineers, quantity surveyors, and structural and mechanical engineers. Based on their understanding and observations, respondents assigned numerical values to these aspects: 5 for very high, 4 for high, 3 for normal, 2 for small, and 1 for nil or very little. Previous research has used this scale[58, 73-77].
3.3 population and sampling method
Considering the novelty of the concept of sustainability in Egypt, stratified sampling was used to investigate its effects on a subset of the population. In addition, the sample size used in this study was decided by a purpose analysis of the methodology [78]. 95 out of 150 people were contacted for this study, with a response rate of around 70%, by Yin [79] recommendation that a small sample size is ideal for doing PLS-SEM. According to the findings of these studies, this rate of return is satisfactory [80, 81].
3.4 Validity and Reliability analysis
Quality and efficiency in the analysis are of utmost importance, so it's important to have a well-thought-out theory framework, as well as acceptable protocols and methodologies, before beginning any investigation. Research findings are trusted because of reliability and validity, as stated by Neuman [82]. Testing for reliability is often performed to rule out biases and flaws in the analysis and guarantee that the same results will be achieved if the researcher repeats the same technique. This contrasts with validity, which is based on objective evaluations of the study's quality and approximation of the truth. The method of data collection was determined after reliability testing to guarantee repeatable results. However, face validity and content validity are the two conventional measures of validity used to evaluate whether the components of the survey instrument are appropriate and reflective of the intended research design [83]. To ensure the reliability of the research instrument, a sample of 12 research professionals from the field and the academy were chosen at random. This was done by the suggestion made by Sushil and Verma [84], that face validity is checked by having experts assess the contents to test them and ensure they appear sufficient. Experts analyze the substance of the test to ensure the items seem adequate (face validity). To determine the validity and reliability of the findings, Cronbach's alpha was utilized. The reliability of the scale was assessed by examining the correlations among the items in the sample to determine whether they were indeed related. Values of Cronbach's alpha above.8 are considered very reliable, while values between.6 and.7 are considered adequate. Since each of the alpha Cronbach values is more than that, this means that they are all suitable. Six, the average set correlation was greater than 0.3, suggesting that there were stable internal variables present in all objects [85].
3.5 Analytical approach (PLS-SEM)
The study's fundamental purpose is to investigate what elements contribute to long-term sustainability in delivery using SEM as a forecasting tool. To meet the needs of the research, the SEM technique describes the relationship between as many observable and unobservable variables as necessary. According to Amaratunga, et al. [86], SEM is an effective method for addressing the presence of errors in variables. The study used the structural equation modelling technique to create a model and determine the connection between elements related to sustainable delivery. To fill the gap left by the lack of familiarity with hypothesis testing procedures, Byrne [87] argued that structural equation modelling (SEM) has become a standard non-experimental research tool. Also, using research articles from the MIS Quarterly, Ringle, et al. [88] concluded that this technique has improved with time. Yuan, et al. [89] also found that SEM is a common and widely used method for analyzing social data. The SEM method was utilized in the study since it is widely known to be effective in the construction industry [90]. Additionally, the activities in the sustainable delivery elements have been evaluated using a PLS model that includes both reflective and formative variables.
However, in this paper, the methodology are not arranged. For instance, I can not understand the purpose of these sentences " To better understand what goes into making a building project sustainable, four models were reviewed from the literature and compared to the best option made possible by sustainable delivery. Multiple Linear Regressions, Structural Equation Modeling, System Dynamic, and Artificial Neural Networks are the methods considered (ANN). This is because there is a connection between the variables that has prevented us from using the regression equation. In a major way, this constrains the applicability of the regression equation [81]. System dynamics could not be used because the data utilized in the study had no temporal context". What is the purpose of reviewing of these models. Are the ANN is used in the study. The authors are not accurate when they describe their work. The methodology is the core of the research. Thus, it needs to be more clarified and arranged better with a major focus on the study. This is a critical point and needs to be considered.
We are grateful to the reviewer comment, which we fully agree with him. This part has been investigated and unwanted statements and tools have been deleted. Now the current form describes directly focus on the study.
The language of the paper needs more improvements.
Many thanks for the editor's valuable comment. The whole language of the paper has been enhanced according to the reviewer's comments
Round 3
Reviewer 1 Report
Most of the comments have been realized. However, the language of the paper must be revised.
Author Response
Thank you very much for reviewing our manuscript. We also greatly appreciate the will complimentary comments and suggestions.
The whole language of the paper has been enhanced according to the reviewer's comments